# Genetic Diversity and Relationships of *Listeria monocytogenes* Serogroup IIa Isolated in Poland

**DOI:** 10.3390/microorganisms10030532

**Published:** 2022-02-28

**Authors:** Beata Lachtara, Kinga Wieczorek, Jacek Osek

**Affiliations:** Department of Hygiene of Food of Animal Origin, National Veterinary Research Institute, 24-100 Pulawy, Poland; beata.lachtara@piwet.pulawy.pl (B.L.); kinga.wieczorek@piwet.pulawy.pl (K.W.)

**Keywords:** *Listeria monocytogenes*, WGS, serogroup IIa, food, molecular typing

## Abstract

In the present study, 100 *L. monocytogenes* isolates of serogroup IIa from food and food production environments in Poland were characterized towards the presence of virulence, resistance, and stress response genes using whole-genome sequencing (WGS). The strains were also molecularly typed and compared with multi-locus sequence typing (MLST) and core genome MLST analyses. The present isolates were grouped into 6 sublineages (SLs), with the most prevalent SL155 (33 isolates), SL121 (32 isolates), and SL8 (28 isolates) and classified into six clonal complexes, with the most prevalent CC155 (33 strains), CC121 (32 isolates), and CC8 (28 strains). Furthermore, the strains were grouped to eight sequence types, with the most prevalent ST155 (33 strains), ST121 (30 isolates), and ST8 (28; strains) followed by 60 cgMLST types (CTs). WGS data showed the presence of several virulence genes or putative molecular markers playing a role in pathogenesis of listeriosis and involved in survival of *L. monocytogenes* in adverse environmental conditions. Some of the present strains were molecularly closely related to *L. monocytogenes* previously isolated in Poland. The results of the study showed that food and food production environments may be a source of *L. monocytogenes* of serogroup IIa with pathogenic potential.

## 1. Introduction

*Listeria monocytogenes* is responsible for a foodborne disease in humans called listeriosis, which is characterized by a high mortality rate [1,2,3,4]. According to the recent European Food Safety Authority (EFSA) and European Centre for Disease Prevention and Control (ECDC) report, in 2020, a total of 1876 confirmed listeriosis cases in humans in the European Union, with the notification rate of 0.42 per 100,000 population, were noted [5]. Among them, 62 infections (0.16 notification rate) were identified in Poland.

*L. monocytogenes* is widespread in the environment, including food production areas, and many different kinds of foods were linked to human infection [6,7].

*L. monocytogenes* is classified into four evolutionary lines (I, II, III, and IV) and four molecular serogroups (IIa, IIc, IIb, and IVb), which cover different serotypes (1/2a, 3a, 1/2c, 3c, 1/2b, 3b, 4b, 4d, and 4e) [8,9,10]. Several studies have indicated strains divergence regarding their ability to persist in the environment as well as their virulence potential [10]. It has been shown that among isolates classified into the evolutionary line II, there are *L. monocytogenes* of serogroup IIa, which are often over-represented in food and food processing environments; such isolates are usually characterized by a high prevalence of various virulence marker genes and are often identified in humans with listeriosis [11,12,13,14].

Recent application of whole-genome sequencing (WGS) into bacterial molecular characterization provides much data on the relationship of *L. monocytogenes* from different sources and allows the classification of the strains into evolutionary lines and genetic types [15,16,17]. Based on the WGS, the multilocus sequence typing (MLST) approach allows classification of *L. monocytogenes* into clonal complexes (CCs) and sequence types (STs), which are linked with persistence of the strains in food, food production environments or with a high potential to cause listeriosis [18,19,20,21]. It has been shown that *L. monocytogenes* of CC121 classified as serogroup IIa has been often isolated from different food processing plants for several years, which makes them serious problems for the food industry and poses a potential threat for consumers [20]. Other data also suggest that *L. monocytogenes* IIa serogroup, classified to CC8, CC155, and other clonal complexes, have persistent properties and have been identified in different niches or sources [15,17,18]. WGS data also provides an opportunity to characterize *L. monocytogenes* based on the core genome multilocus sequence typing (cgMLST) that allows a standardized comparison of the sequences of the tested strains with the publicly available genomes present in databases [11,22,23].

The WGS analysis also delivers information related to the virulence properties and potential of *L. monocytogenes* to survive under a wide range of suboptimal environmental conditions. One of such markers is Premature Stop Codon Mutations (PMSC) in the *inlA* gene responsible for the production of internalin A, which mediates bacterial adhesion and invasion of epithelial cells in the human intestine [24]. Strains with such truncated genes are rarely involved in human listeriosis infections but more often are detected among food isolates [25]. There are also reports showing that persistent isolates from food processing plants and ecosystems exhibited higher resistance to quaternary ammonium compounds (QACs) than non-persistent ones, especially strains classified to IIa serogroup [26,27]. Furthermore, QAC-tolerant *L. monocytogenes* have a higher ability to form protective biofilms and are characterized by an increased tolerance to other types of QAC-based biocides used in the food industry [28].

The aims of the present study were: (i) to characterize the virulence potential and assess the genetic diversity of *L. monocytogenes* serogroup IIa isolated in Poland using WGS analysis; (ii) to determine the molecular relationships of isolates tested; and (iii) to compare the present *L. monocytogenes* with the sequences of other isolates of IIa serogroup isolated previously in Poland available in BIGSdb-Lm database.

## 2. Materials and Methods

### 2.1. Sample Collection

A total of 1439 *L. monocytogenes* isolates were obtained during routine food and food production environments microbiological investigations performed by official veterinary laboratories located in 13 out of 16 administrative regions (voivodeships) of Poland during 2013–2019 using the ISO-11290-1 standard [29]. The isolates were then sent to the National Veterinary Research Institute, National Reference Laboratory for *L. monocytogenes* in Pulawy and tested towards molecular serogroups with PCR as described [8,30]. For the present study, 100 *L. monocytogenes* isolates classified to serogroup IIa and recovered from raw meat (*n* = 19), ready-to-eat (RTE) food of animal origin (*n* = 48), and from food production environments (FPEs) (*n* = 33) were selected and stored in Viabank (Medical Wire and Equipment, Corsham, Wiltshire, UK) at −80 °C. Detailed information on the *L. monocytogenes* isolates used is shown in Appendix A.

### 2.2. DNA Isolation and Sequencing

*L. monocytogenes* was cultured on tryptone soya-yeast extract agar at 37 °C for 18–24 h and a loopful of bacteria was transferred into 100 µL of TRIS (Tris-(hydroxymethyl)-aminomethane) buffer (A&A Biotechnology, Gdynia, Poland). DNA was extracted using the Genomic Mini protocol (A&A Biotechnology) modified by adding 20 µL of lysozyme (10 mg/mL; Sigma-Aldrich, St. Louis, MO, USA) and incubation of the samples at 37 °C for 30 min.

DNA quality and concentration were measured by NanoDrop or Qubit 3 (Thermo Fisher Scientific, Waltham, MA, USA), and sequencing libraries were prepared with a Nextera XT DNA Sample Preparation Kit (Illumina, San Diego, CA, USA) and a KAPA HyperPlus Kit (Hoffman-La Roche, Basel, Switzerland) according to the manufacturer’s instruction. The libraries were then sequenced in a MiSeq (Illumina) and NextSeq (Illumina) with approximately 50× and 100× average coverage. The sequences were trimmed using Trimmomatic v.0.36 [31] and Fastp v.0.22.0 and finally assembled wit SPAdes v.3.9.0 and v.3.15.3 [32]. 

The detailed *L. monocytogenes* sequence parameters used in the present study are listed in Appendix A. All genome sequences obtained were deposited in the Listeria PasteurMLST database (https://bigsdb.pasteur.fr/listeria (accessed on 15 January 2022)) under the accession numbers 79375-79474.

### 2.3. WGS Analysis

All *L. monocytogenes* sequences were analyzed using the publicly available web-based WGS tools on the BIGSdb-Lm platform (https://bigsdb.pasteur.fr/listeria (accessed on 15 January 2022)) [11,22,33].

MLST profiles with the same alleles for 7 loci were classified into sequence types (ST) and grouped into clonal complexes (CCs) if at least 5 out of 7 loci were the same as previously described [11,34]. cgMLST (1748 loci) profiles were grouped into cgMLST types (CTs) and sublineages (SLs), using the cut-offs of 7 and 150 allelic mismatches, respectively, as previously described [11]. Minimum spanning tree (MST) was generated using BioNumerics software version 7.6 (Applied Maths, Sint-Martens-Latem, Belgium) based on the cgMLST allele number and the predefined template for categorical data.

Assemblies were also screened in silico for virulence factor, antimicrobial, metal, and biocide resistance genes, Listeria Stress Islands as well as the *sigB* and rhamnose operons using the BIGSdb-Lm platform [11,19,35,36,37,38]. The obtained sequences were further analyzed for the presence of *qacH* and *emrC* gene using BLAST (https://blast.ncbi.nlm.nih.gov/Blast.cgi (accessed on 20 January 2022)) and the reference sequences described before [39,40].

The putative prophage determinants within the genomes of the *L. monocytogenes* isolates were identified using the PHASTER (PHAge Search Tool Enhanced Release) web server [41,42]. To identify the presence of plasmid sequences the WGS sequences were analyzed with PlasmidFinder software 2.1 for the specified Gram-positive scheme (https://cge.cbs.dtu.dk/services/PlasmidFinder (accessed on 18 January 2022)) [43].

### 2.4. Comparison of L. monocytogenes Sequences

The WGS sequences of the isolates tested in the present study were compared with 39 sequences of Polish *L. monocytogenes* publicly available in BIGSdb-Lm database. Isolates classified to CC8, CC121, and CC155, recovered from humans, food, and food production environments were selected. The cgMLST profiles of all compared strains were created using sequences of the 1748 loci according to the scheme described before [11]. The minimum spanning trees (MSTs) based on cgMLST profiles were constructed using the BioNumerics 7.6 software as previously described [30]. The strains were classified to the same cluster when less than 7 allelic differences were identified. Phylogenetic tree was constructed using BioNumerics 7.6 based on cluster analysis (similarity matrix) of the categorical differences in the allelic cgMLST profiles for each isolate using the single linkage calculating method. Detailed information on the *L. monocytogenes* isolates used for comparisons are listed in Appendix A.

## 3. Results and Discussion

### 3.1. WGS-Based Typing of L. monocytogenes of IIa Serogroup

Analysis of WGS data of the 100 *L. monocytogenes* isolates allowed to classify them into 6 clonal complexes (Figure 1). Three main CCs identified covered the vast majority of the isolates, i.e., CC155 (33; 33.5%), CC121 (32; 32.0%), and CC8 (28; 28.0%), and they were originated from all sources tested. The remaining seven strains, mainly recovered from RTE food, were grouped into three CCs, with single isolates in each (Figure 1). *L. monocytogenes* of the CCs identified in the present study were also previously detected among 100 strains of IIa serogroup recovered from food but were also often isolated from humans with listeriosis [12,14,18,44,45,46,47].

Further analysis of WGS sequences revealed that all 100 current isolates were classified into eight sequence types, with the most prevalent ST155 (33; 33.0%), ST121 (30; 30.0%), and ST8 (28; 28.0%) (Figure 2). The remaining nine strains belonged to five distinct STs (ST7, ST12, ST37, ST451, and ST1398) and were mainly isolated from RTE food. The number of STs in relation to different sources of the strains tested is shown in Table 1. Three STs (ST8, ST121, and ST155) were identified in *L. monocytogenes* isolated from all three sources analyzed (Table 1). The results of other authors also showed that ST155 and ST121 were often identified among *L. monocytogenes* of food origin [14,15,21]. It seems that such isolates may have a molecular background that allows them to survive in food and food production environments in different areas for a long time [11,15].

Based on the WGS sequences, the present isolates were grouped into 6 sublineages (SLs), with the most prevalent SL155 (33 isolates), SL121 (32 isolates), and SL8 (28 isolates) (Appendix A). During the investigation of Hurley et al. (2019), some of their 100 *L. monocytogenes* strains were classified to the same sublineages as identified in the present study: SL121 (12% isolates), SL8 (10%), SL7 (8%), SL451 (4%), and SL37 (1%). Our previous analysis of 48 *L. monocytogenes* isolates originated from food and food processing environments in Poland showed 25 different CTs grouped into seven SLs, but none of those strains was classified into the same sublineages as identified in the current study. However, those isolates were classified to other serogroups [48].

The present isolates were further grouped into 60 different cgMLST types (CTs), with the most numerous CT1170 (17 isolates) and CT750 (12 strains), both mainly recovered from RTE food and food production environments (a total of 12 and 11 isolates, respectively), but also from raw meat (five isolates of CT1170) (Figure 3). There were also CTs unique to the source of isolation (Appendix A). The number of CTs identified in the strains of different origins tested is shown in Table 1.

A similar study performed by Hurley et al. (2019) showed a large molecular diversity among 100 *L. monocytogenes* isolates from food and food processing environments in Ireland. These authors identified a total of 37 distinct cgMLST types, with the most abundant being CT1526 (20 strains), followed by CT1844 (11 isolates) and CT1828 (7 strains). However, none of these CTs was detected in the current investigation.

### 3.2. General Molecular Characteristics of L. monocytogenes

Several genetic molecular markers involved in the pathogenesis of *L. monocytogenes* infection and survival of the bacteria in adverse environmental conditions were identified from the WGS data of 100 isolates tested (Appendix A). One of these important genes are four Listeria Pathogenicity Islands (LIPIs). Among them, LIPI-1 are essential for invasion, intracellular growth, and further spread to adjacent cells during the listeriosis infectious cycle [49]. The LIPI-1 island was identified in all isolates tested in the present study, irrespective of their clonal complex (Appendix A). It has been previously shown that this gene is present in all *L. monocytogenes* and is composed of six genes, including *prfA*, *actA*, *hly*, *mpl*, *iap*, *plcA*, and *plcB* (Table 2) [50]. On the other hand, none of the *L. monocytogenes* harbored other LIPI virulence islands, i.e., LIPI-2 involved in the expression of invasion-associated surface proteins, LIPI-3 with the *llsX* gene responsible for the production of listeriosin S (LLS toxin), which promotes post-translational modifications, and LIPI-4, a cluster of six genes and is involved in neural and placental infection [33,51]. The lack of these genes may suggest that the currently tested isolates had a low pathogenic potential for humans.

Other important genes playing a possible role in the persistence of *L. monocytogenes* in food production environments, such as sequences encoding resistance to cadmium, e.g., the *cadA1* gene located on plasmid-borne *Tn5422* transposon and a metal-responsive transcriptional repressor *cadC*, previously identified, e.g., among isolates of CC155 [52], were not detected in any of the strains tested, including those belonging to CC155 (Appendix A). This result is opposite to the study of Wagner et al. (2020), who showed that the vast majority of their *L. monocytogenes* CC155 isolates tested these cadmium resistance markers. Another investigation demonstrated that the *cadA1* gene was identified mainly in strains of serotypes 1/2a and 1/2b originated from food and food-processing environment [53,54]. In the present study, all *L. monocytogenes* tested were classified to serogroup IIa, which covers serotype 1/2a, but all isolates were negative for this sequence.

The WGS data were also analyzed towards the presence of sigma factor protein (*sigB*) operon, responsible for the adaptation of *L. monocytogenes* to several stress conditions, which may be present in food production environments, such as low temperatures, pH, high hydrostatic pressure, and biofilm formation [55,56]. It was shown that the *sigB* operon was present in all currently investigated strains (Appendix A), and similar results were previously described by other authors [14,21,57].

Analysis of WGS sequences of 100 *L. monocytogenes* isolates tested showed the presence of several mobile genetic elements such as plasmids (Appendix A). Among them, the most common was plasmid pLM5578 identified only among all 32 strains classified to CC121. It has been previously shown that this plasmid contains sequences encoding resistance to cadmium (*cadA* and *cadC*), which play a role in the survival of such isolated in adverse food processing environments and increase their pathogenic potential [35,58,59]. However, none of the current strains tested with plasmid pLM5578 was positive for the *cadA* and *cadC* sequences (Appendix A).

Other genetic elements which potentially enhance the survival, virulence, or fitness of *L. monocytogenes* are prophages [21,60]. The present investigation revealed the presence of 11 different intact prophage sequences detected among 64 of 100 isolates tested, with the most common PHAGE_Lister_LP_HM00113468_NC_049900 sequence found in 20 strains, mostly of CC8 (Appendix A). Another frequently identified phage sequence was PHAGE_Lister_LP_101_NC_024387 (18 isolates), which was detected among strains classified into five CCs, mainly to CC121 (10 isolates) (Appendix A). Previous studies also showed that *L. monocytogenes* originated from humans with listeriosis but also from food and food production environments from different countries had various prophage sequences inserted into the genome, including those identified in the present study [59,61,62,63]. It may suggest that the phage sequences presently detected are commonly found among *L. monocytogenes* identified all over the world. However, their exact role in survival, persistence, and virulence has to be further evaluated [60,64,65].

The remaining virulence, resistance, stress response, and other genes identified in 100 *L. monocytogenes* isolates tested are discussed in paragraphs related to particular clonal complexes and shown in Table 2, Appendix A. Additionally, the phylogenetic tree based on cgMLST profiles with the presence/absence of the most important genes of interest as well as additional information about all isolates tested is shown in Appendix A.

### 3.3. Molecular Characteristics of L. monocytogenes of Different CCs

#### 3.3.1. CC155 Isolates

*L. monocytogenes* of the most prevalent clonal complex (CC155; *n* = 33) harbored only isolates classified into one MLST sequence type ST155 and nine different cgMLST types, with the most prevalent CT1170 (17; 51.5% isolates) followed by CT9831 (6; 18.2%) (Appendix A). Strains of CC155 were mainly recovered from raw meat (14; 42.4% isolates) and FPEs (13; 39.4%). As described by Wagner et al. (2020) *L. monocytogenes* of CC155 are often identified in food and food production environments as well as among clinical isolates due to their genetic attributes supporting their ability to persist in environments as well as to infect humans. One of these features, especially important in surviving under adverse environmental conditions, is resistance to quaternary ammonium compounds (QAC) such as benzalkonium chloride (BC), which depends on the presence of efflux pump genes, such as *bcrABC* cassette [66]. In the present study, 12 out of 33 (36.4%) *L. monocytogenes* of CC155 were positive for this marker. The isolates were mainly from raw meat (7 isolates) and food production environments (4 strains) (Appendix A). Furthermore, none of the strains possessed the *qacA* gene, which is also involved in BC resistance [66]. A study by Wagner et al. (2020) showed that all but one of 20 *L. monocytogenes* classified to CC155, isolated from food and clinical sources, were positive for the *bcrABC* cassette. Since, in most cases, the *qacA* gene is located on the pLM80 plasmid, it can be easily lost or transferred between diverse *L. monocytogenes* of various sources [58,67].

Further analysis of WGS sequences of *L. monocytogenes* CC155 tested revealed that none of them was positive for the chromosomally located *Tn6188_qac* (*ermC*) transposon harboring the *qacH* efflux pump gene involved in tolerance to BC and other QACs [68]. Similar observations were made by Wagner et al. (2020), who showed that none of their strains tested possessed this marker. On the other hand, a study by Meier et al. (2017) demonstrated that the majority of Swiss and Finnish *L. monocytogenes* clinical and food isolates resistant to BC were *qacH*-positive [26].

A correlation between the ability to biofilm formation by different *L. monocytogenes* strains and the presence of stress survival islet 1 (SSI-1) was previously demonstrated [69]. Genomic analysis of the current isolates classified into CC155 showed that all of them were positive for this marker (Table 2). It has been previously described that such SSI-1 positive *L. monocytogenes* are able to form stronger biofilm structures compared to SSI-1 negative strains. Thus, they show better persistent properties, which allow them to survive in food production environments for a long time [70,71].

Another important marker involved in biofilm formation is the *inlA* gene responsible for the expression of internalin A [72]. It has been demonstrated that *L. monocytogenes* carrying the PMSC mutation in the *inlA* gene, which results in the reduced length of InlA protein, showed enhanced biofilm-forming abilities but decreased virulence compared to the isolates that had full-length InlA [25]. Furthermore, such mutation occurs more commonly among food isolates than in strains responsible for human infections [73]. In the current investigation, none of the CC155 *L. monocytogenes* tested possessed the truncated *inlA* internalin gene, which may suggest their increased pathogenic potential for humans (Appendix A).

#### 3.3.2. CC121 Isolates

*L. monocytogenes* classified to clonal complex CC121 (*n* = 32) belonged to two STs: ST121 (30 isolates) and ST1398 (2 strains), among them, a total of 28 different cgMLST types were identified (Appendix A). The majority of the CC121 isolates were originated from RTE food (20; 62.5% strains). The remaining isolates were either from food production environments (8; 25.0%) or raw meat (4; 12.5%). *L. monocytogenes* classified to sequence type ST121 seems to be very well adapted to conditions present in food production environments and was isolated during several studies related to food production [59,74,75]. Furthermore, such isolates have been identified in any of the hypo-virulent clones but are sometimes associated with human listeriosis [26,59,76].

Screening of the WGS data towards pathogenic markers among *L. monocytogenes* CC121 revealed that the *bcrABC* and *qacA* genes connected with resistance to BC were not identified in any of the 32 strains (Appendix A). These results are opposite to other studies where such BC tolerance gene markers were present, at least in some strains of clonal complex 121 [26,77].

Further analysis of the WGS data was performed towards another efflux pump gene responsible for the increased tolerance of *L. monocytogenes* to QAC, i.e., *Tn6188_qac (ermC)* transposon with the *qacH* efflux pump gene connected with pLMST6 plasmid [78]. It was found that this transposon was present in 13 (40.6%) strains of CC121 (Table 2). This BC resistance sequence was also often identified during other studies on *L. monocytogenes* isolated from food or food production environments [26,71]. However, there are also studies that demonstrated that this efflux pump determinant was not identified in any of *L. monocytogenes* isolates of food and human origin; however, these strains were classified into ST155 type [21].

The investigation of the presence of stress survival islet 1 (SSI-1), one of the gene markers involved in biofilm formation and persistence of *L. monocytogenes* showed that, in contrast to strains of CC155, all CC121 isolates tested were negative for this sequence (Table 2). This may suggest that the CC121 *L. monocytogenes* tested do not have strong properties responsible for the survival of the bacteria in adverse food production environment conditions [70,71]. However, the current CC121 isolates were positive for stress survival islet 2 (SSI-2) involved in the survival of *L. monocytogenes* under alkaline and oxidative stress conditions [36].

Analysis of the present WGS data towards the *inlA* internalin trait, another gene involved in biofilm development by *L. monocytogenes*, including the PMSC sequence mutation, revealed that all but one isolates of CC121 tested carried the truncated gene (Appendix A), which resulted in the reduced length of InlA protein and had an influence on enhanced biofilm-forming abilities [25]. This finally makes such isolates being well adapted for survival and persistence in the environment present in food production environments [15,25,77]. On the other hand, such isolates may be less pathogenic for humans [73].

#### 3.3.3. CC8 Isolates

All 28 *L. monocytogenes* isolates tested classified into clonal complex CC8 belonged to one sequence type (ST8) and 17 different cgMLST types, with the most predominant CT750 (12; 42.9% isolates) (Appendix A). All but one strain were recovered from RTE food (17; 60.7%) and food production environments (10; 35.7%). *L. monocytogenes* of ST8, such as the characterized above ST121, was previously often isolated from food and food production areas, which suggests that strains of such sequence type can survive in adverse conditions present in such environments [13,74,77,79,80]. Furthermore, such isolates have probably increased virulence properties since they were also responsible for human listeriosis cases [47,59,81,82].

WGS data analysis of CC8 strains towards virulence and resistance markers showed that the main genes responsible for resistance to benzalkonium chloride, i.e., *bcrABC* and *qacA*, were not present in any of the 28 isolates tested. This result is identical to the current *L. monocytogenes* classified to CC121. Only two CC8 isolates of cgMLST types CT750 and CT9837, both of food production environments origin, were positive for the *Tn6188_qac* (*ermC*) transposon involved in QAC and other various disinfectants resistance widely used in food production (Table 2) [68,71]. Isolates belonging to CC8 were also previously identified in food and in humans suffering from listeriosis [18,26,45,46,47,83]. However, there are also studies in which none of the *L. monocytogenes* classified to CC8 was positive for the *Tn6188_qac* (*ermC*) marker [49,80].

The ability of biofilm formation, another important characteristic of persistent *L. monocytogenes* isolates responsible for food contamination, depends on several molecular markers such as stress survival islet 1 (SSI-1) and *inlA*. The analysis of WGS sequences of CC8 strains tested revealed that the SSI-1 gene was present in all isolates (Table 2). Previous studies suggested serotype-specific differences in biofilm development linked to the presence of SSI-1 [69,84,85]. Additionally, all but one isolates of CC8 were positive for the full length of the *inlA* (internalin) gene (Appendix A). This may indicate that such strains demonstrated weaker biofilm-forming abilities but enhanced virulence compared to the isolates that had PMSC mutation in this sequence as currently identified in *L. monocytogenes* classified to CC121 (Appendix A) [25].

#### 3.3.4. Isolates of the Remaining CCs

A total of seven remaining *L. monocytogenes* isolates tested were classified to CC7 (4 strains), CC11 (2 isolates), and CC37 (one strain). They were recovered from RTE food or food production environments and characterized towards sequence and cgMLST types (Appendix A). The isolates showed a high molecular diversity belonging to four STs (ST7; ST12, ST37, and ST451) and six CTs (CT720; CT798; CT4399; CT9808; CT9849; CT9850). *L. monocytogenes* of such clonal complexes were previously isolated from food, although their prevalence was rather low [86,87].

Analysis of the WGS sequences towards the main genes responsible for resistance to biocides such as benzalkonium chloride, i.e., *bcrABC* and *qacA*, revealed that none of these markers was identified (Appendix A). Furthermore, all strains were negative for the *Tn6188_qac* (*ermC*) transposon involved in QAC and other disinfectants resistance used in food production [68].

All these seven *L. monocytogenes* tested did not possess sequences encoding resistance to cadmium (*cadA1* gene). However, the stress survival islet 1 (SSI-1) involved in biofilm formation was present in four isolates classified to ST7 and ST12 (two strains of each), whereas another biofilm-related marker (*inlA* gene) was found in all isolates, but it was without the PMSC mutation suggesting their weaker biofilm-forming ability [25].

### 3.4. Molecular Relationships of the Current and Other Polish L. monocytogenes

*L. monocytogenes* of CC155 were grouped into six clusters covering from two to nine isolates (Appendix A). The most numerous group of *L. monocytogenes* consisting of isolates classified to the same cgMLST type (CT1170; *n* = 17) was separated into three distinct clusters and were obtained from eight different voivodeships during years 2014–2019 (Appendix A). Among one cluster with a total of nine out of 17 CT1170 isolates, there were strains obtained during years 2014–2018 in seven voivodeships and were mainly originated from RTE meat (Appendix A). The remaining eight CT1170 *L. monocytogenes* isolates were classified into two clusters and were originated from wielkopolskie (five strains) and warmińsko-mazurskie (three strains) voivodeships, respectively, and were recovered from 2017 to 2019 (Appendix A).

Furthermore, among CC155 isolates, there were two clusters with the molecularly identical strains classified into the same cgMLST types: CT9831 (six isolates, all of raw meat origin), identified in the same voivodeship (mazowieckie) during years 2013–2014 (Appendix A) and CT9832 (three isolates from two voivodeships; Appendix A), respectively. The other two strains of CC155 belonged to CT9829 and were isolated from FPEs in two voivodeships in 2015 and 2018 (Appendix A).

Molecular relationship of the current *L. monocytogenes* isolates classified into CC155, based on the cgMLST analysis, are shown in Figure 4. They were also compared with six other strains of the same clonal complex previously identified in Poland. MST analysis showed a close relationship of one isolate (ID79401) of CT1170, originated from food, with the strain of ID46913 also isolated from food (only five allelic differences). Other Polish *L. monocytogenes* sequences available in BIGSdb-Lm database were not closely related to the isolates investigated during the present study (Figure 4).

Comparative analysis of *L. monocytogenes* strains belonging to CC121 and showing over 95% similarity based on the cgMLST data revealed three clusters with two or three strains in each, classified into three different CTs (Appendix A). The strains were either isolated from the same voivodeship (łódzkie; three isolates of CT9852 from 2013 of RTE meat origin; Appendix A or kujawsko-pomorskie, two isolates of CT9821, from 2015 from RTE and 2016 from FPEs, Appendix A) or from different voivodeships (łódzkie i podkarpackie, both strains of CT903 from 2014 from RTE and 2015 from raw meat, Appendix A).

Molecular relationship of the current *L. monocytogenes* isolates classified into CC121, based on the cgMLST analysis, are shown in Figure 5. They were also compared with 19 other strains of the same clonal complex previously identified in Poland. The analysis revealed that only one presently tested strain (ID79402, classified to CT1249 and isolated from food production environments in kujawsko-pomorskie voivodeship in 2015) showed a close relationship (three allelic differences) with two strains (ID46896 and ID46897) originated from the same source in 2019. The remaining 31 current *L. monocytogenes* CC121 isolates have not displayed a close relationship with previously isolated strains identified in our country, i.e., showed more than seven allelic differences (Figure 5).

Analysis of molecular similarity of strains classified to CC8 revealed that the most predominant *L. monocytogenes* of CT750 (12 isolates) were grouped into four clusters containing from four to two identical or closely related isolates (Appendix A). For example, isolate ID79412 recovered in 2016 from food production environments in małopolskie voivodeship displayed from three to four allelic differences with two other isolates (ID79386 and ID79389) recovered from two voivodeships (łódzkie in 2014 and podkarpackie in 2015), and both strains were of RTE meat origin (Appendix A). Strains with ID79449 and ID79408, also recovered from food production environments but in different voivodeships and during different years (Appendix A), were also closely related as tested by the cgMLST profiles (five alleles mismatches; Figure 6). Furthermore, strain ID79412 was very similar to *L. monocytogenes* ID34379 isolated in Poland in 2012 from a person with listeriosis. However, the remaining sequences of 11 historical Polish strains did not show a close molecular relationship with the current isolates belonging to CC8 (Figure 6).

Analysis of genetic similarity of the remaining *L. monocytogenes* strains classified to CC7, CC11, and CC37 showed that only one currently tested isolate (ID79454 of CC11, recovered from RTE food in 2018 in warmińsko-mazurskie voivodeship) showed a three allelic difference with one strain (ID41692) isolated in 2015 from a person suffering from listeriosis.

## 4. Conclusions

WGS analysis of *L. monocytogenes* classified to serogroup IIa, isolated from food and food production environments in Poland, revealed a prevalence of several resistance and virulence genes among the isolates tested. Identification of certain genetic traits, especially those encoding resistance to disinfectants and responsible for biofilm formation as well as isolation of the strains with the same molecular profiles in different years from the same geographical areas, suggest that at least some of the *L. monocytogenes* tested possess the ability to persist in food production environments for a long time. The investigated isolates showed a high genotypic diversity as identified during the analysis of the WGS data, especially based on the cgMLST sequences. A close molecular relationship of the current *L. monocytogenes* isolates tested with those recovered previously from similar food and food production sources and from patients with listeriosis indicate that foods and their production environments may be a potential source of pathogenic strains for humans.

## Figures and Tables

**Figure 1 microorganisms-10-00532-f001:**
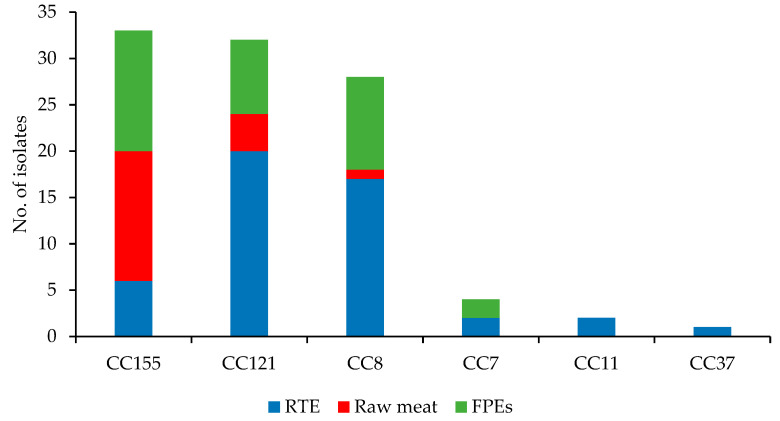
Distribution of clonal complexes (CCs) among of 100 *L. monocytogenes* isolates tested (RTE, ready-to-eat meat; FPEs, food production environments).

**Figure 2 microorganisms-10-00532-f002:**
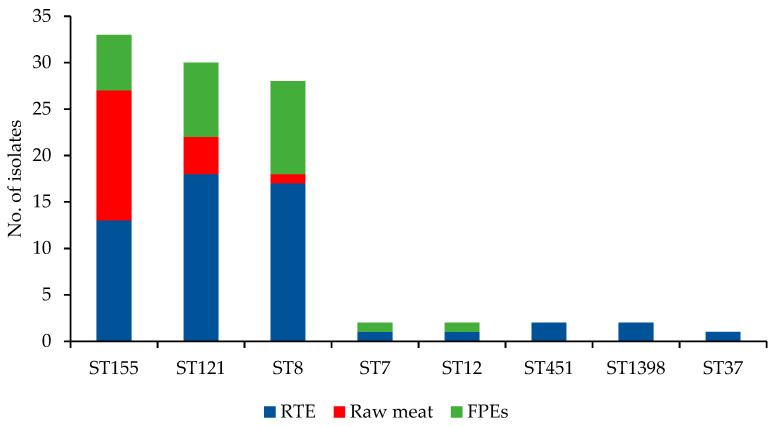
Distribution of sequences types (STs) among of 100 *L. monocytogenes* isolates tested (RTE, ready-to-eat meat; FPEs, food production environments).

**Figure 3 microorganisms-10-00532-f003:**
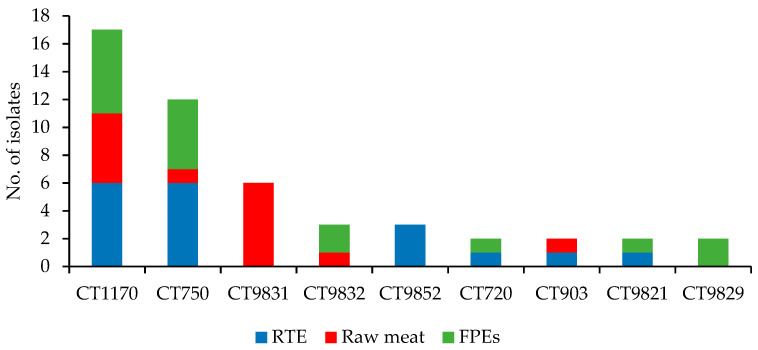
Distribution of cgMLST types (CTs) among of 100 *L. monocytogenes* isolates tested, which were identified in more than one isolate.

**Figure 4 microorganisms-10-00532-f004:**
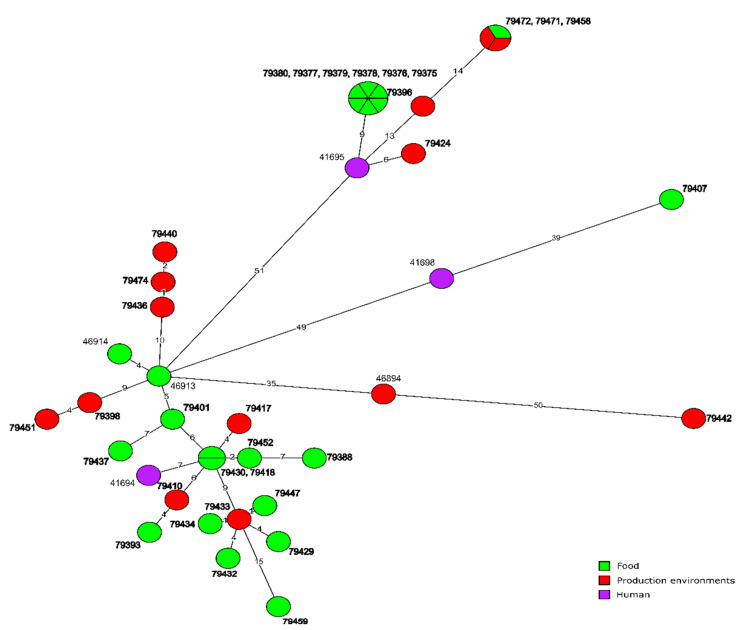
Minimum spanning tree (MST) based on the cgMLST profiles of *L. monocytogenes* CC155 tested in the present study with publicly available six strains of CC155 from Poland. The Source of the isolates are represented by colored circles where the size is proportional to the number of strains. Numbers on the branches show alleles differences between neighboring nodes (CTs). The strain numbers in bold represent *L. monocytogenes* isolates from the present study.

**Figure 5 microorganisms-10-00532-f005:**
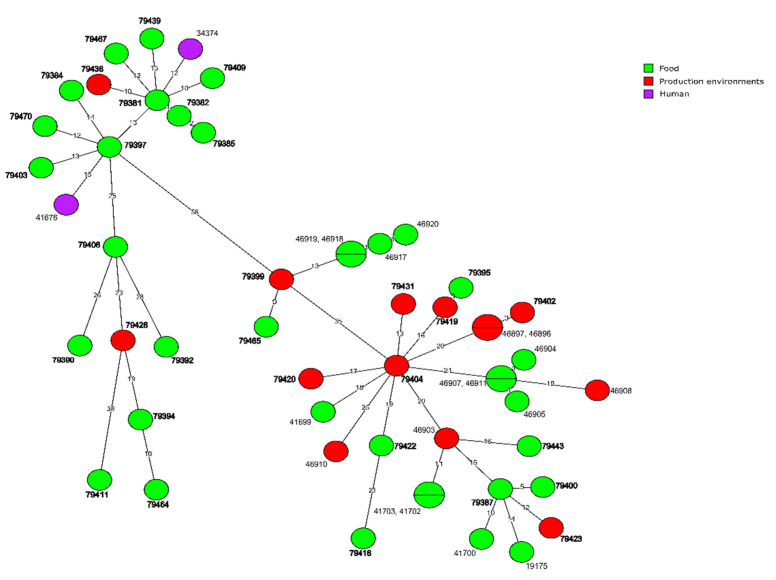
Minimum spanning tree (MST) based on the cgMLST profiles of *L. monocytogenes* CC121 tested in the present study with publicly available 20 strains of CC121 from Poland. Source of the isolates are represented by colored circles where the size is proportional to the number of strains. Numbers on the branches show alleles differences between neighboring nodes (CTs). The strain numbers in bold represent *L. monocytogenes* isolates from the present study.

**Figure 6 microorganisms-10-00532-f006:**
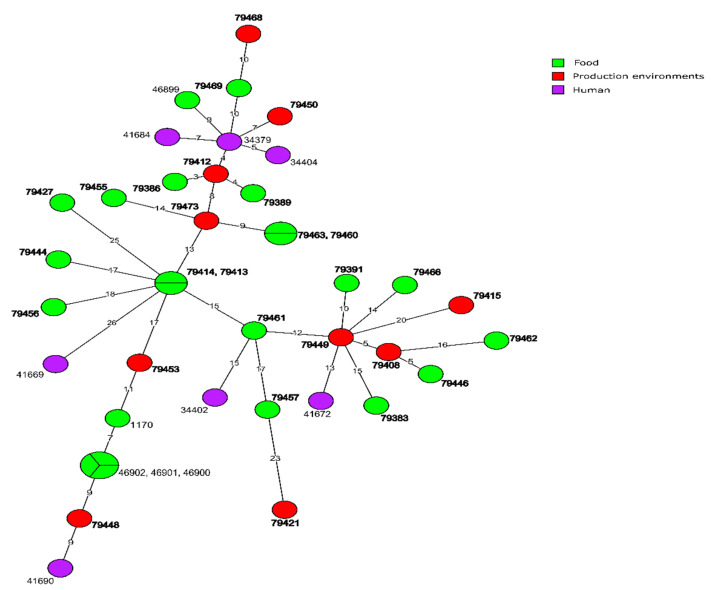
Minimum spanning tree (MST) based on the cgMLST profiles of *L. monocytogenes* CC8 tested in the present study with publicly available twelve strains of CC8 from Poland. Source of the isolates are represented by colored circles where the size is proportional to the number of strains. Numbers on the branches show alleles differences between neighboring nodes (CTs). The strain numbers in bold represent *L. monocytogenes* isolates from the present study.

**Table 1 microorganisms-10-00532-t001:** Prevalence of sequence (ST) and cgMLST (CT) types among *L. monocytogenes* tested.

Source of Isolates	No. of Total Types		Most Common Types (No. of Isolates)		No. of Types Unique for the Source		Common to All Sources	
	ST	CT	ST	CT	ST	CT	ST	CT
Raw meat (*n* = 19)	3	10	ST155 (14)	CT1170 (6)	0	6	ST8; ST121; ST155	CT750; CT1170
RTE food (*n* = 48) ^a^	8	36	ST121 (18)	CT1170 (6)	3	31		
FPEs (*n* = 33) ^b^	5	22	ST155 (13)	CT9831 (6)	0	17		

^a^ RTE, ready-to-eat. ^b^ FPEs, food production environments.

**Table 2 microorganisms-10-00532-t002:** Distribution of virulence and resistance factor genes among *L. monocytogenes* isolates tested in relation to the clonal complexes (CCs) and sequence types (STs).

Trait	Gene	No. of Isolates			
		CC155 (*n* = 33)	CC121 (*n* = 32)		CC8 (*n* = 28)
		ST155 (*n* = 33)	ST121 (*n* = 30)	ST1398 (*n* = 2)	ST8 (*n* = 28)
Metal & disinfectants resistance	*bcrA*	12	0	0	0
	*bcrB*	12	0	0	0
	*bcrC*	12	0	0	0
	*ermC (Tn6118_qac)*	0	13	0	2
Stress Islands	SSI1_lmo0444	33	0	0	27
	SSI1_lmo0445	33	0	0	28
	SSI1_lmo0446	33	0	0	28
	SSI1_lmo0447	33	0	0	28
	SSI1_lmo0448	32	0	0	28
	SSI2_lin0464	0	30	2	0
	SSI2_lin0465	0	30	2	0
Internalins	*inlA*	33	29	2	27
	*inlB*	33	30	2	28
	*inlC*	33	30	2	28
	*inlE*	33	30	2	28
	*inlF*	33	0	0	28
	*inlG*	33	0	0	28
	*inlH*	33	30	2	28
	*inlJ*	33	30	2	27
	*inlK*	33	30	2	27
LIPI-1	*prfA*	33	30	2	28
	*plcA*	33	27	2	28
	*hly*	33	30	2	28
	*mpl*	33	29	2	27
	*actA*	33	30	2	28
	*plcB*	33	30	2	26

## Data Availability

The genome sequences obtained in the present study were deposited in the Listeria PasteurMLST database (https://bigsdb.pasteur.fr/listeria (accessed on 15 January 2022)) under the accession numbers 79375-79474.

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
