# Peer review of "Genetic Diversity and Relationships of Listeria monocytogenes Serogroup IIa Isolated in Poland"

_microorganisms, 2022, doi:10.3390/microorganisms10030532_

Round 1

Reviewer 1 Report

The article is well written, the methodology is robust. The methods and the results are extensively described. The results are compared to the current literature in the field.

As there is a huge amount of data, especially for the presence/absence of genes of interest, my only concern is that the article suffer of a lack of one or more synthetic figure. e.g Phylogenetic trees annotated with the genes presence /absence, the origin of the isolates, the geographical information… could help the readers to have a more global overview of these really interesting data. These kind of figures could be easily performed using Itol or phandango software.

Author Response

Thank you very much for the critical reading and careful evaluation of our manuscript. According to your comments, we have prepared an additional figure with the presence/absence of the most important genes of interest as well as additional informational about all isolates tested (source and time of isolation, molecular classification, etc.). This figure was created with the BioNumerics software and is marked as Figure S1 of the supplementary file (please find attached). The information on this figure has been included in the text of the manuscript (marked in green).

Reviewer 2 Report

This article aimed: (i) to characterize the virulence potential and assess the genetic diversity of L. monocytogenes serogroup IIa isolated in Poland using WGS analysis; (ii) to determine the molecular relationships of isolates tested; and (iii) to compare the present L. monocytogenes with the sequences of other isolates of IIa serogroup isolated previously in Poland available in BIGSdb-Lm database. The authors concluded that food and food production environments may be a source of L. monocytogenes of serogroup IIa with pathogenic potential.

The article is straightforward, and it contains original information. 

Minor edits shown below are recommended: 

Line 31. Revise to “were” and clarify “0.16 rate.”

Line 101. Revise to “with.”

Line 150. Revise to “among 100 ….”

Line 171. Italicize to “L. monocytogenes.”

Line 184. Revise to “is.”

Lines 234 and 494. Revise to “current.”

Line 307. Revise to “identified in any of ….”

Lines 319-323. The sentence is confusing, and rephrase the sentence in two sentences.

Author Response

Thank you very much for the critical reading and careful evaluation of our manuscript. According to your comments, we have revised the manuscript and made the modifications marked in red in the revised version of the manuscript.

Comment: Line 31. Revise to “were” and clarify “0.16 rate.”

Response: Done

Comment: Line 101. Revise to “with.”

Response: Done

Comment: Line 150. Revise to “among 100 ….”

Response: Done

Comment: Line 171. Italicize to “L. monocytogenes.”

Response: Done

Comment: Line 184. Revise to “is.”

Response: Done

Comment: Lines 234 and 494. Revise to “current.”

Response: Done

Comment: Line 307. Revise to “identified in any of ….”

Response: Done

Comment: Lines 319-323. The sentence is confusing, and rephrase the sentence in two sentences.

Response: This long sentence has been revised and split into two shorter sentences.